# Discovery of Eicosapentaenoic Acid Esters of Hydroxy Fatty Acids as Potent Nrf2 Activators

**DOI:** 10.3390/antiox9050397

**Published:** 2020-05-08

**Authors:** Siddabasave Gowda B. Gowda, Hirotoshi Fuda, Takayuki Tsukui, Hitoshi Chiba, Shu-Ping Hui

**Affiliations:** 1Faculty of Health Sciences, Hokkaido University, Kita-12, Nishi-5, Kita-Ku, Sapporo 060-0812, Japan; siddabasavegowda.bommegowda@hs.hokudai.ac.jp (S.G.B.G.); hirofuda@gmail.com (H.F.); 2Department of Nutrition, Sapporo University of Health Sciences, Nakanuma Nishi-4-3-1-15, Higashi-Ku, Sapporo 007-0894, Japan; tsukuitk@gmail.com (T.T.); chibahit@med.hokudai.ac.jp (H.C.)

**Keywords:** antioxidants, EPA, lipid synthesis, lipid biochemistry, lipid mediators, lipid droplet, reporter gene assay, Nrf2

## Abstract

Branched fatty acid esters of hydroxy fatty acids (FAHFAs) are a recently discovered class of biologically active lipids with anti-inflammatory and anti-diabetic properties. Despite the possible link between endogenous FAHFA levels and nuclear factor erythroid 2-related factor 2 (Nrf2), their possible function as antioxidants and the mechanisms involved in this are unknown. Here, we investigate FAHFAs’ plausible antioxidant potential with reference to their effect on the Nrf2 levels, oxidative stress, and lipid droplet oxidation in human hepatocytes (C3A). Six authentic FAHFAs were chemically synthesized and performed activity-based screening by reporter gene assay. Among them, eicosapentaenoic acid (EPA) esterified 12-hydroxy stearic acid (12-HSA) and 12-hydroxy oleic acid (12-HOA) FAHFAs showed less cytotoxicity compared to their free fatty acids and potent activators of Nrf2. To define their mode of action, relative levels of nuclear Nrf2 were determined, which found a higher amount of Nrf2 in nucleus of cells treated with 12-EPAHSA compared to the control. Furthermore, 12-EPAHSA increased the expression of Nrf2-dependent antioxidant enzyme genes (*NQO1*, *GCLM*, *GCLC*, *SOD-1*, and *HO-1*). Fluorescence imaging analysis of linoleic-acid-induced lipid droplets (LDs) in C3A cells treated with 12-EPAHSA revealed the strong inhibition of small-size LD oxidation. These results suggest that EPA-derived FAHFAs as a new class of lipids with less cytotoxicity, and strong Nrf2 activators with plausible antioxidant effects via the induction of cytoprotective proteins against oxidative stress, induced cellular damage.

## 1. Introduction

Fatty acid esters of hydroxy fatty acids (FAHFAs) are a new class of endogenous lipids with beneficial biological effects, including antidiabetic and anti-inflammatory properties [1]. To date, their biosynthesis and metabolism are not well elucidated [2]. They are found in blood and many tissues, such as adipose tissue, liver, heart, kidneys, and pancreas in mammals, and have been identified in several food materials [3]. In particular, palmitic acid esters of hydroxy stearic acid (PAHSAs) are the major esters in human and mice. PAHSAs received considerable attention because their levels were significantly decreased in serum of insulin-resistant patients [4], breast cancer patients [5], and the milk of obese women compared to lean mothers [6], as well as suppressing inflammatory markers [7], decreasing T-cell activation in ulcerative colitis [8], and improving glucose tolerance by stimulating insulin secretion via GPR120 signaling [1]. Supplementation of ω-3 fatty acids such as docosahexaenoic acid (DHA) in diabetic patients and obese mice increased the circulating levels of DHA esterified hydroxy linoleic acid FAHFAs and plays a protective role [9]. 

Cellular defense against reactive oxygen species (ROS) is mediated by direct and indirect antioxidants. Direct antioxidants are often small molecules, which scavenge ROS through redox reactions, whereas indirect antioxidants induce a series of antioxidant enzymes by Nrf2 activation and catalyze ROS breakdown in cells [10]. Under normal conditions, Nrf2 is constantly degraded in a Keap1-dependent manner via the ubiquitin-proteasome pathway [11]. However, the presence of oxidants and electrophiles could lead to the disruption of Keap1-Nrf2 interaction [12,13]. Recently, there is growing interest in promising Nrf2 activators because many physiological studies showed the Nrf2 activation alleviated pathological conditions, including chronic kidney disease, pulmonary hypertension, neurodegenerative disorder, and cancer [14,15]. In other words, the activation of Nrf2 could lead to the upregulation of antioxidant enzymes and protect cells against oxidative stress-induced damages. Despite the diverse functions of FAHFAs, studies revealing their potential antioxidant activity have not been reported. A recent study showed that the Nrf2-mediated antioxidant defense mechanism may be linked to the biosynthesis of PAHSAs [16], however there are no studies on the evaluation of the intrinsic antioxidant capabilities of FAHFAs. In this study, we aim to explore the plausible antioxidant potential of novel synthetic fatty acid esters of 12-hydroxy stearic acid (12-HSA), 12-hydroxy oleic acid (12-HOA) and their possible mode of action in the Nrf2 defense mechanism and lipid droplet oxidation in cultured human hepatocytes by reporter gene assay and fluorescence imaging analysis. We also interrogated their possible biosynthesis in liver cells.

## 2. Materials and Methods

12-hydroxystearic acid, methyl ricinoleate, oleoyl chloride, linoleoyl chloride, dry pyridine, anhydrous dichloromethane, CDCl_3_, and 4-dimethylaminopyridine were purchased from Tokyo Chemical Industry (Tokyo, Japan). Dicyclohexyl carbodiimide, lithium hydroxide, Methanol for LC/MS, and all other reagents of synthetic grade were obtained from Wako Pure Chemical Corporation (Tokyo, Japan). Docosahexaenoic acid and Eicosapentaenoic acid with purity ≥98%, were purchased from Cayman Chemicals (Ann Arbor, MA, USA). Thin-layer chromatography (TLC) Silica gel 60G F_254_ glass plates (20 × 20 cm) were obtained from Merck (Tokyo, Japan) and spots were visualized by spraying 5% methanolic H_2_SO_4_. The column chromatography was performed by using spherical type Silica Gel N60 of particle size 40–50 μm, obtained from Kanto Chemical Industry (Tokyo, Japan). ^1^H and ^13^C NMR spectra were acquired using 400 MHz JNM-ECX400P (JEOL, Japan). The spectra were processed using ACD/NMR software and the chemical shifts(*δ*) values are given in ppm. The accurate mass measurements were performed by LTQ Orbitrap XL (Thermo Fisher scientific). Low-resolution electrospray ionization mass spectra (LR-ESI-MS) was recorded by LXQ (Thermo Fisher scientific), The methyl ester of 12-hydroxystearic acid (12-HSA) was prepared by direct methylation 12-HSA by refluxing at 80 °C in methanolic 2N HCl.

### 2.1. General Procedure for Synthesis of Oleoyl and Linoleoyl Esters of 12-Hydroxy Fatty Acids

To a solution of 12-HSA-OMe or 12-hydroxyoleic acid methyl ester (12-HOA-OMe) (100 mg, 0.318 mmol) in dry dichloromethane (4 mL) and dry pyridine (125.6 μL, 5 equivalents (eq)) at 0 °C, oleoyl chloride or linoleoyl chloride (1.1 eq) was added slowly and stirred for 24 h at room temperature under nitrogen atmosphere. Reaction mixture was extracted with dichloromethane by successive washing with 0.5 M HCl, saturated NaHCO_3_ and NaCl, and dried over sodium sulfate. The crude extract was concentrated under reduced pressure and the resulting mixture was purified with flash column chromatography (hexane/ethyl acetate = 99/1~95/5) to give methyl esters of 12-OAHSA, 12-LAHSA, 12-OAHOA, and 12-LAHOA with a yield of about 65–70%. Further, their methyl esters dissolved in THF and were subjected to saponification using 1M aqueous LiOH (2 eq) at room temperature for a period of 24 h. Then, the reaction was neutralized with 2N HCl under an ice bath, followed by extraction with diethyl ether, concentrated in vacuum and purified via silica gel column chromatography using hexane/ethyl acetate/AcOH (93/6/1) to give 12-OAHSA (**2**), 12-LAHSA (**3**), 12-OAHOA (**2a**), and 12-LAHOA (**3a**), FAHFAs with a yield ranging between 65% and 70%. 

#### General Procedure for Synthesis of Eicosapentaenoic Acid Esters of 12-Hydroxy Fatty Acids

To a solution of 12-HSA-OMe or 12-hydroxyoleic acid methyl ester (12-HOA-OMe) (200 mg, 0.63 mmol) in dry dichloromethane (15 mL), DMAP (38.7 mg, 0.5 eq) and DCC (327.5 mg, 2.5 eq) at 0 °C under nitrogen atmosphere, EPA (264.7 µL,1.3 eq) was added slowly and stirred for 24 h at room temperature under nitrogen atmosphere. The reaction mixture was extracted with dichloromethane by successive washing with 0.5M HCl, saturated NaHCO_3_, NaCl and dried over sodium sulfate. The crude extract was concentrated under reduced pressure and the resulting mixture was purified with flash column chromatography (hexane/ethyl acetate = 98/2~90/10) to give methyl esters of 12-EPAHSA, and 12-EPAHOA with a yield of about 60–65%. Further, their methyl esters were subjected to alkali hydrolysis as described above and the reaction was neutralized with 2N HCl under an ice bath, followed by extraction with ethyl acetate, concentrated in vacuum and purified via silica gel column chromatography (hexane/ethyl acetate/AcOH = 95/5/0~80/20/1) to give 12-EPAHSA (**4**) and 12-EPAHOA (**4a**), FAHFAs with a yield ranging between 70% and 75%. 

### 2.2. Cell Culture and Assays

The cell culture and assays were performed by the previously published report with minor modifications [17]. The C3A cells (derivative of Human HepG2, CRL-10741) were purchased from ATCC (Manassas, VA, USA) and kept at 37 °C in an incubator under a humidified atmosphere of 5% CO_2_ using minimum essential medium (MEM) with 10% (*v/v*) fetal bovine serum (FBS), and 1% Penicillin–Streptomycin–Neomycin mixture (all were obtained from Thermo Fisher Scientific, Japan). The cell cytotoxicity is evaluated using CCK-8 assay kit (Dojindo Molecular Technologies). The Graph Pad Prism 6.03 software was used to determine the half-maximal inhibitory concentration (IC_50_) of each FAHFAs. The reporter gene assay was performed according to the previously established protocol in our laboratory [17). At first, cultured C3A cells were seeded into 96-well plates and incubated for 24 h, followed by transfecting the cells using the FuGENE HD Transfection Reagent. The vectors, pGL4.37[luc2p/ARE/Hygro] and pGL4.75[hRluc/CMV] (Promega) of mass ration 20:1, were used to co-transfect the cells and incubated for 24 h. Then, the transfection reagent was washed off, and pre-dissolved FAHFAs in MEM was applied to the cells and incubated for 24 h. Dual-Glo Luciferase Assay System (Promega) was used to measure the luciferase activity in control and FAHFA-treated samples. The fold change in relative luciferase activity was calculated by dividing the fluorescence intensity in FAHFAs treated samples to that of control.

#### 2.2.1. Nuclear Protein Extraction and Western Blot Analysis

The nuclear Nrf2 accumulation was determined by the nuclear protein extraction kit and Western blotting analysis. Cells were incubated with the FAHFAs in MEM without FBS for 24 h. Nuclear protein was fractionated using a Nuclear Extraction Kit (Active Motif). Samples (10 µg protein/well) were electrophoresed in NuPAGE 4–12% Bis-Tris Gels and transferred to Immobilon-P Membranes. The membranes were soaked in a solution of 5% dry milk in TBS containing 0.05% (*v/v*) Tween-20. Then, the membranes were exposed to either rabbit anti-Nrf2 antibodies (1:5,000, ab31163; Abcam, Cambridge, UK) or rabbit anti-Lamin B1 antibodies (1:50,000, ab16048; Abcam, Cambridge, UK) overnight. After washing the membranes in tris-buffered saline (with 0.05% Tween-20), the membrane was incubated for 2 h after treatment with anti-rabbit IgG (horseradish peroxidase-labelled) antibodies (1:10,000). The signals obtained were identified using a ChemiDoc MP Imaging System and LumiGLO Chemiluminescent Substrate System (Bio-Rad Laboratories Inc., Tokyo, Japan), and the bands were quantified by ImageJ software. The relative Nrf2 level was determined by taking the ratios of intensity of Nrf2 band to Lamin B1 band, and data were expressed by increase in fold change compared to control.

#### 2.2.2. First-Strand cDNA Synthesis and Real-Time PCR

Approximately 4.0 × 10^5^/well of cultured C3A cells were seeded into 24-well plates. After 24 h of incubation, the FAHFAs dissolved in MEM were added to the cells and further incubated for 24 h. The cells were harvested by scraping and total RNA was extracted using ReliaPrep RNA Cell Miniprep System (Promega). Extracted RNA samples were treated with DNAse and GoScript Reverse Transcription assay System (Promega, Japan) was used for synthesis of cDNA by the procedures established earlier in our laboratory [17]. The relative expression of Nrf2 target genes was determined using real-time PCR (RT-PCR). 

The PCR mixture contained 10 µL of SsoFast EvaGreen Supermix (Bio-Rad Laboratories Inc.), 1:5-diluted cDNA (2 µM) and 0.5 µM of each primer in a total volume of 20 µL. The reaction cycle comprised a holding stage of 30 s at 95 °C, followed by denaturation cycles of 5 s at 60 °C and 10 s extension at 72 °C. The amount of PCR product formed is confirmed by the fluorescence observed at the end of each cycles using CFX Connect (Bio-Rad Laboratories Inc.). The expression level of each target gene was normalized by glyceraldehyde-3-phosphate dehydrogenase (GAPDH). The primer sequences of the target genes, heme oxygenase-1 (HO-1), NAD(P)H:quinone oxidoreductase 1 (NQO1), glutamate–cysteine ligase modulatory subunit (GCLM), glutamate–cysteine ligase catalytic subunit (GCLC), catalase (CAT), and superoxide dismutase 1(SOD1) used in the study are identical to that of our earlier report [17].

### 2.3. Fluorescent Imaging Analysis

Fluorescent imaging of lipid droplet (LD) and oxidized lipid droplet (oxLD) was performed according to a previously established method in our laboratory [18]. Briefly, the cells were precultured in 0.1% gelatin-coated glass bottom dish with 6 × 10^5^ cells in 3 mL of the medium for 24 h. Then, the cells were treated with 400 µM linoleic acid (LA) with 0–125 µM 12-EPAHSA. After 8 h incubation, the cells were stained using 5 µM SRfluor 680-phenyl (Funakoshi co. Ltd., Tokyo, Japan) a fluorescent probe for neutral lipids, 10 µM Liperfluo, and 10 µg/mL Hoechst33342 (Dojindo laboratories, Kumamoto, Japan) a probe for lipid peroxides and nuclei, for 30 min at 37 °C. Fluorescence was recorded using the BZ-9000 Keyence fluorescence microscope having filter sets; excitation: 360/40 nm, emission: 460/50 nm, dichroic mirror: 400 nm (blue); excitation: 470/40 nm, emission: 525/50 nm, dichroic mirror: 495 nm (green); excitation: 620/60 nm, emission: 700/75 nm, dichroic mirror: 660 nm (red). Acquired images were processed by ImageJ 1.50i software. The intersectional images were obtained from binarized images of bright field, SRfluor and Liperfluo in the same visual field. Statistical analyses were performed using the GraphPad Prism 6 software. Data derived from three or more replicates are shown as mean values ± SEM. Student’s *t-*test (two-tailed) was used to study statistically significant differences between the groups and a *p*-value of <0.05 was considered as statistically significant.

## 3. Results

The synthesis of fatty acid esters of 12-hydroxy stearic acid (12-HSA) and 12-hydroxy oleic acid(12-HOA) is described in Scheme 1. The methyl esters of the respective fatty acids were subjected to acylation reaction under two different conditions. The FAHFAs of oleic acid (OA) and linolenic acid (LA) were synthesized by direct acylation using respective acyl chlorides, followed by mild alkali hydrolysis using lithium hydroxide to afford 12-oleic acid hydroxy stearic acid (12-OAHSA (**2**)), 12-linoleic acid hydroxy stearic acid (12-LAHSA (**3**)), 12-oleic acid hydroxy oleic acid (12-OAHOA (**2a**)), and 12-linolenic acid hydroxy oleic acid (12-LAHOA (**3a**)). 12- eicosapentaenoic acid hydroxy stearic acid (12-EPAHSA (**4**)), and 12-eicosapentaenoic acid hydroxy oleic acid (12-EPAHOA (**4a**)) were synthesized by direct coupling of EPA with methyl esters of 12-HSA and 12-HOA, followed by mild alkali hydrolysis. The ^1^H-NMR and ^13^C-NMR chemical shifts for each compound are provided below and their spectra are given in the Appendix A.

12-OAHSA **(2):** R_f_ = 0.25 (hexane/ethyl acetate = 7/1); ^1^H-NMR (400 MHz, CDCl_3_) *δ* 5.36–5.33 (m, 2 H), 4.90–4.84 (m, 1 H), 2.35 (t, 2H, *J* = 7.3,15.1 Hz), 2.28 (t, 2H, *J* = 7.3,15.1 Hz), 2.03–1.98 (m, 4 H), 1.65–1.59 (m, 4 H), 1.51–1.50 (m, 4 H), 1.30–1.26 (m, 42 H), 0.90–0.86 (m, 6 H). ^13^C-NMR (100 MHz, CDCl_3_) *δ* 180.08, 174.06, 130.28, 130.05, 74.42, 35.04, 34.47, 34.32, 32.21, 32.06, 30.07, 30.01, 29.82, 29.78, 29.69, 29.62, 29.50, 29.47, 29.44, 29.35, 27.52, 27.47, 25.61, 25.58, 25.47, 24.97, 22.98, 22.87, 14.40, 14.36. The HRMS calculated for C_36_H_67_O_4_ [M-H]^−^, 563.50448, found 563.50334 (−2.02 ppm). 

12-OAHOA **(2a):** R_f_ = 0.25 (hexane/ethyl acetate = 7/1); ^1^H-NMR (400 MHz, CDCl_3_
*δ* 5.49–5.29 (m, 4 H), 4.90–4.87 (m, 1H), 2.36–2.24 (m, 6H), 2.1-1.99 (m, 6 H), 1.67–1.60 (m, 4 H), 1.54–1.52 (m, 2 H), 1.31–1.26 (m, 36 H), 0.90–0.0.86 (m, 6 H); ^13^C-NMR (100 MHz, CDCl_3_) *δ*180.10, 173.93, 132.79, 130.27, 130.04, 124.65,74.01, 34.98, 34.31, 33.93, 32.28, 32.20, 32.04, 30.06, 30.00, 29.80, 29.61, 29.49, 29.43, 29.37, 29.31, 27.61, 27.51, 27.46, 27.44, 25.64, 25.41, 24.95, 22.97, 22.86, 14.39, 14.34. HR-ESIMS calculated for C_36_H_65_O_4_ [M-H]^−^,561.48883, found 561.48771 (−1.99 ppm).

12-LAHSA **(3):** R_f_ = 0.25 (hexane/ethyl acetate = 7/1); ^1^H-NMR (400 MHz, CDCl_3_) *δ* 5.40–5.31 (m, 4 H), 4.88–4.85 (m, 1 H), 2.75 (t, 2H, *J* = 6.8,13.3 Hz), 2.32 (t, 2H, *J* = 7.3,15.1 Hz), 2.26 (t, 2H, *J* = 7.8,15.1 Hz), 2.07–2.02 (m, 4 H), 1.67–1.59 (m, 4 H), 1.51–1.50 (m, 4 H), 1.30–1.26 (m, 36 H), 0.91–0.86 (m, 6 H). ^13^C-NMR (100 MHz, CDCl_3_) *δ*179.87, 173.92, 132.80, 130.52, 128.22, 124.66, 74.03, 34.99, 34.27, 33.95, 32.29, 32.05, 31.83, 29.92, 29.81, 29.65, 29.45, 29.38, 29.33, 29.31, 27.62, 27.50, 27.48, 25.93, 25.65, 25.41, 24.96, 22.98, 22.87, 14.40, 14.36. HR-ESIMS calculated for C_36_H_65_O_4_ [M-H]^−^, 561.48883, found 561.48786 (−1.7 ppm).

12-LAHOA **(3a):** R_f_ = 0.25 (hexane/ethyl acetate = 7/1); ^1^H-NMR (400 MHz, CDCl_3_) *δ* 5.48–5.31 (m, 6 H), 4.90–4.85 (m, 1 H), 2.37–2.25 (m, 6H) 2.08–1.99 (m, 6H), 1.65–1.60 (m, 4 H), 1.54–1.51 (m, 2 H), 1.38–1.26 (m, 32 H), 0.91–0.86 (m, 6 H). ^13^C-NMR (100 MHz, CDCl_3_) *δ*179.87, HR-ESIMS calculated for C_36_H_63_O_4_ [M-H]^−^, 559.47318, found 559.47279 (−0.69 ppm).

12-EPAHSA **(4):** R_f_ = 0.2 (hexane/ethyl acetate = 5/1); ^1^H-NMR (400 MHz, CDCl_3_) *δ* 5.41–5.30 (m, 10 H), 4.90–4.84 (m, 1 H), 2.86–2.80 (8H, m), 2.36–2.28 (m, 4H), 2.14–2.06 (m, 4 H), 1.74–1.59 (m, 4 H), 1.51–1.50 (m, 4 H), 1.26 (m, 22 H), 0.98 (t, 3H, J = 7.3, 15.1 Hz), 0.88 (t, 3H, J = 6.9, 13.7 Hz). ^13^C-NMR (100 MHz, CDCl_3_) *δ*179.49, 173.54, 132.10, 129.15, 128.80, 128.65, 128.33, 128.19, 127.96, 127.10, 74.33, 34.22, 34.04, 31.82, 29.56, 29.46, 29.28, 29.12, 26.74, 25.70, 25.62, 25.41, 25.37, 25.11, 24.75, 22.66, 20.63, 14.34, 14.13. HR-ESIMS calculated for C_38_H_63_O_4_ [M-H]^−^, 583.47318, found 583.47218 (−1.7 ppm).

12-EPAHOA **(4a):** R_f_ = 0.2 (hexane/ethyl acetate = 5/1); ^1^H-NMR (400 MHz, CDCl_3_) *δ* 5.47–5.30 (m, 12 H), 4.90–4.87 (m, 1 H), 2.84–2.81 (8H, m), 2.36–2.27 (6H, m), 2.11–2.01 (m, 6H), 1.73–1.61 (m, 4 H), 1.31–1.26 (m, 18 H), 0.99–0.95 (m, 3H), 0.89–0.86 (m, 3H). ^13^C-NMR (100 MHz, CDCl_3_) *δ*179.65, 173.33, 132.51, 132.0, 130.19, 129.04, 129.0, 128.71, 128.54, 128.23,128.14, 128.08, 128.06,127.86, 127.63, 127.0, 124.28, 73.82, 34.06, 33.97, 33.60, 33.32, 31.94, 31.71, 29.48, 29.11, 29.05, 28.99, 27.29, 26.61, 26.44, 25.60, 25.52, 25.33, 24.95, 24.64, 22.55, 20.53, 14.23, 14.02. HR-ESIMS calculated for C_38_H_61_O_4_ [M-H]^−^,581.45753, found 581.45655 (−1.6 ppm).

The cytotoxicity of all the synthetic FAHFAs and authentic hydroxy fatty acids were evaluated against human hepatoma cells (C3A) by CCK-8 assay, and the values of representative species in mean ± SD (*n* = 6) are given as, 12-OAHSA (499 ± 2.9), 12-LAHOA (673 ± 1.7), 12-EPAHSA (415 ± 2.5), 12-EPAHOA (358 ± 2.5), EPA (117 ± 2.5), and 12-HSA (161 ± 2.7), respectively. The cell viability logarithmic plots for compounds **4** and **4a** are provided in Figure 1A. The results show that FAHFAs are relatively less toxic when compared to their respective free fatty acids. The ability of each FAHFA at its non-cytotoxic concentrations regarding the activation of Nrf2 was examined with a reporter gene assay using Dual-Glo Luciferase Reporter Assay System (Promega), in which the antioxidant response element (ARE) drove the transcription of the luciferase reporter gene according to previously established protocol in our lab, with minor modifications [17].

Among all the screened compounds **2**, **2a**, **3**, and **3a** did not shows any significant activity against Nrf2 activation (Appendix A). In other words, EPA-derived compounds such as **4**, and **4a** shows the activation of Nrf2 in a dose-dependent manner (Figure 1B). The relative luciferase activity drastically increased to more than twenty-fold at a concentration of 250 µM of each compound **4** and **4a**. Further, the effect of the compound 12-EPAHSA (**4**) on accumulation of Nrf2 protein in nuclear fraction was examined using nuclear protein extraction kit, and Western blotting analysis, and the results are shown in Figure 1C, D. The 12-EPAHSA shows the significantly increased levels of nuclear Nrf2 with increasing concentration.

Next, the relative expression of Nrf2 target antioxidant genes in the samples derived was determined using real-time PCR and the data of significantly altered genes are shown in Figure 1E. The 12-EPAHSA significantly increased the expression levels of Nrf2-targeted cytoprotective genes, such as *NQO1, GCLM, GCLC,* and *SOD-1,* in a dose-dependent manner. The expression levels of *CAT,* and *HO-1,* showed the increasing tendency with change in concentration. The levels of GAPDH were used as an internal control in each experiment. The 12-EPAHSA treatment significantly reduced the oxidative stress biomarker, DDIT3/CHOP relative expression levels, however the effect was not concentration-dependent (Appendix A). Furthermore, the antioxidant potential of 12-EPAHSA was evaluated by fluorescent imaging by treating it with lipid droplet (LD) and oxidized lipid droplet (oxLD) by the method established earlier in our laboratory [18]. After 8-h incubation, small and a few large LDs were formed and oxidized in the linoleic (LA)-treated C3A cells (Figure 2A). The oxidized LDs were significantly reduced with increasing concentrations of 12-EPAHSA treatment. The number of total LDs were unchanged (Figure 2B1). On the other hand, the number of small oxLDs significantly decreased in a dose-dependent manner (Figure 2B2). The degree of oxidation in small LDs was also showed a decreasing trend (Figure 2B3) in response to 12-EPAHSA treatment when compared to untreated cells. The LDs detected were not only the neutral lipids; the lipid hydroperoxides were also stained with green fluorescent probe, which reduced significantly with the 12-EPAHSA treatment.

Next, to examine the whether C3A cells could synthesize FAHFAs, we added the equimolar concentrations of EPA and 12-HSA to the C3A cells and incubated for 12 h at 37 °C. Then, the cells were washed with phosphate-buffered saline, and extracted the total lipids (~1 × 10^5^ cells) for LC-MS/MS analysis. Figure 3 shows the schematic representation of 12-EPAHSA biosynthesis, with the concentration-dependent production of 12-EPAHSA in C3A cells. The concentrations of each fatty acids is less than their IC_50_ values (i.e., 16 and 32 µM). The results show 6-fold higher levels of 12-EPAHSA, at 32 µM, compared to 16 µM treated samples.

## 4. Discussion

Lipids are signaling molecules with roles in the membrane structurer and sources of energy. However, recently discovered classes of lipids such as FAHFAs are known to protect diabetes and inflammation via the activation of G-protein coupled receptors [1]. These lipids are found in adequate amounts in mammalian tissues such as adipose tissue, liver, heart, and daily foods [3]. Despite their structural diversity and the biological activity of these endogenous lipids, the antioxidant potential through the induction of the Nrf2 activation pathway is unrevealed. Nrf2 is a transcription factor that regulates the gene expression of antioxidant and enzymes. In this research, we synthesized and screened 12-HSA, 12 HOA-derived FAHFAs against Nrf2 activation and found that EPA esterified FAHFAs increased the Nrf2-transactivation in a dose-dependent fashion compared to untreated cells (Figure 1B). 12-EPAHSA (**4**) led to a considerable increase in nuclear Nrf2 protein levels (Figure 1C,D). These results are concurrent with a previous study, which showed that free n-3 fatty acids such as EPA and DHA can protect the astrocytes against oxidative stress via Nrf2-dependent signaling [19]. Nrf2 translocation is one of the key events required for the regulation of Keap1-Nrf2 pathway and it is considered as evidence of the activation of the system. Our analysis results showed the accumulation of higher amount of Nrf2 in the nucleus of cells treated with 12-EPAHSA compared to the control, suggesting the possible activation of nuclear Nrf2.

Besides this, Nrf2 target antioxidant enzymes such as *NQO1, GCLM, GCLC, SOD-1,* and *HO-1* were increased by the 12-EPAHSA treatment (Figure 1E), suggesting that EPA-derived FAHFAs are involved in the upregulation of these antioxidant defense enzymes, possibly via Nrf2 activation, which is evident in the literature [10,17]. Moreover, the knockdown of Nrf1 can increase the expression of antioxidant genes including NQO1, and the activation of NQO1 is induced by Nrf2 much more strongly than by Nrf1Δ30 in a reporter gene assay system [20]. Considering these facts, we assume that the translocated nuclear Nrf2, rather than Nrf1, plays a major role in the FAHFAs-induced expression of antioxidant enzymes. However, the limitation of our study was that the ARE reporter is not specific for Nrf2 but also reports Nrf1 activation [20]. Hence, there is a possibility of the involvement of Nrf1 in antioxidant enzymes’ upregulation upon 12-EPAHSA treatment, which needs further experimental studies such as the specific knockdown of Nrf2 via siRNA. Furthermore, studies are required to confirm the effect of FAHFAs on Nrf2 activation using multiple cell lines. The overall antioxidant effects of 12-EPAHSA were summarized in Figure 4. The activation of antioxidant enzymes and their cytoprotective mechanisms against oxidative stress induced damages by 12-EPAHSA. The cellular transporters of FAHFAs are still unknown, but our results revealed that 12-EPAHSA could activate the nuclear Nrf2 and induce the activation of Nrf2-targeted genes to protect cells against oxidative-stress-related damages. We tested the activities of free fatty acids (EPA and 12-HSA) to know which structural part of 12-EPAHSA is active, and the results suggested that EPA is the most potent activator of Nrf2, but not 12-HSA (Appendix A). However, its activity is much lower than the parent 12-EPAHSA, which is comparatively less toxic than EPA, hence suggesting its possible effect on the cytoplasmic Nrf2 signaling. The antioxidant effect of 12-EPAHSA was evaluated by inducing oxidative stress in HepG2 cells using H_2_O_2_. The results showed the decreasing tendency in oxidative-stress-induced reactive oxygen species levels with increasing concentration of 12-EPAHSA (Appendix A), which further supports its possible action as an antioxidant.

Further, we examined the biosynthesis of FAHFAs in C3A cells by spiking the equimolar concentrations of respective fatty acids (EPA and 12-HSA) at different levels to C3A cells with an incubation period of 12 h, and analyzed the total lipid extracts by LC-MS/MS. The results revealed that FAHFAs are biosynthesized in a dose-dependent manner (Figure 3), indicating the existence of hydroxy fatty acid acyltransferases in the liver cells. However, 12-EPAHSA was not detected in the control samples, suggesting this type of FAHFAs as non-endogenous lipids which may be produced when the fatty acid and hydroxy fatty acid sources are available endogenously. The FAHFAs are assumed to be biosynthesized by acyltransferases in adipocytes, however there is no experimental evidence to confirm their involvement [1]. Our results using cultured C3A cells suggest that FAHFAs could be biosynthesized in liver and have beneficial health effects. Moreover, this type of in vitro technique could be employed for the large-scale production of FAHFAs of biological interest. With our expertise in lipid droplet (LD)-specific imaging analysis, we examined the effect of 12-EPAHSA on linoleic acid-induced LDs and oxidized LDs (including lipid peroxides). Our results showed a significant reduction in several small sizes of oxidized LDs (Figure 2B2), and these results had the same tendency as the change in LDs of DHMBA treated HepG2 cells, suggesting that 12-EPAHSA also has an antioxidant effect on LD oxidation [18]. Knowing the new function of lipid is a critical piece of information needed to understand its endogenous functions. In this study, we have utilized a combination of chemical and biochemical techniques to identify the novel function of FAHFAs as an Nrf2 activator and induced the expression of antioxidant target genes, which could lead to antioxidant effects, which is of great significance for the treatment of oxidative-stress-related disorders.

## 5. Conclusions

In summary, our findings demonstrate that EPA-derived FAHFAs are a novel class of lipids with less cytotoxicity compared to their free fatty acids and potent Nrf2 activators. They also efficiently suppressed the oxidation of small size lipid droplets, oxidative stress induced by H_2_O_2_, and enhanced the gene expressions of cytoprotective antioxidant enzymes. The detailed mechanism involving the biosynthesis, transport of FAHFAs and molecular mechanisms involving Nrf2 activation will of future interest.

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
