# Peer review of "Discovery of Eicosapentaenoic Acid Esters of Hydroxy Fatty Acids as Potent Nrf2 Activators"

_antioxidants, 2020, doi:10.3390/antiox9050397_

Round 1

Reviewer 1 Report

An interesting, meticulously done and properly presented study.

Remarks:

Which concentrations of 12-EPAHSA and 12- EPAHQA can be expected in cells? How the concentrations used are related to these concentrations?

Table  S1: Are the presented data a result of a single experiment? (Lack of SD)

Fig. S1: Which genes are presented?

Fig. S2: Are the concentrations shown on the axis of abscissae sums of concentrations of EPA and 12-HSA or concentrations of single components of the equimolar mixture?

Can the concentrations of 12-EPAHSA synthesized can be estimated on the basis of radioactivity?

Author Response

Response letter

Reviewer 1:

An interesting, meticulously done and properly presented study.

Response: Thank you for your comments.

Remarks:

Which concentrations of 12-EPAHSA and 12- EPAHQA can be expected in cells? How the concentrations used are related to these concentrations?

Response: Thank you for your comments. Originally, we haven’t detected any basal levels of EPA derived FAHFAs in C3A cells as observed in the control experiments (Figure 3). However, spiking the individual fatty acids such as EPA and 12-HSA produced the 12-EPAHSA. The concentration for the experiments were chosen based on their IC50 values.

Table S1: Are the presented data a result of a single experiment? (Lack of SD)

Response: Thank you for your comments. Table S1 is modified as text and move to the main text, as suggested by the other reviewer, and SD values were added. See lines 221-223.

Fig. S1: Which genes are presented?

Response: The legend of figure S1 is modified and term luciferase was added.

Fig. S2: Are the concentrations shown on the axis of abscissae sums of concentrations of EPA and 12-HSA or concentrations of single components of the equimolar mixture?

Response: It corresponds to the “concentrations of single components of the equimolar mixture” and the legend is modified accordingly in the revised MS and moved Figure S2, to main text as Figure 3.

Can the concentrations of 12-EPAHSA synthesized can be estimated on the basis of radioactivity?

Response: Yes, it is possible to estimate the concentrations of 12-EPAHSA by radioactivity, however it requires a suitable-labelled standard.

Reviewer 2 Report

The authors were intended to elucidate effects of branched fatty acid esters of hydroxy fatty acids (FAHFAs), which are endogenous lipids expected to protect tissues from various damages, on cellular mechanisms of Nrf2-mediated antioxidant response. As the result, they found that a FAHFA, eicosapentaenoic acid esterified 12-hydroxy stearic acid 19 (12-EPAHSA), stimulates nuclear accumulation of Nrf2 protein and expression of Nrf2-target genes related to antioxidant response. Based on the results, they insist that FAHFAs are cytoprotective from oxidative stress via activating the master antioxidative transcription factor Nrf2. However, the causal nexus between FAHFAs and the Nrf2 pathway is unclear due to lack of mechanism analyses. To complete this study for publication, additional experiments are strongly advocated as below. Major points (1) Keap1 inactivates Nrf2 while Nrf2 activates antioxidant gene expression. Therefore, “Activation of Keap1-Nrf2 Signaling” and “Keap1-Nrf2 activation” is inaccurate and confusing (In the title; Line 21, 28, 48, 228, 241, 273, 275, and 293). Additionally, this study did not investigate roles of Keap1 at all. These phrases should be replaced with “activation of Nrf2” or “Nrf2 activation”. (2) This study used only one cell line (C3A). However, the effects of 12-EPAHSA must be confirmed by additional experiments using multiple cell lines. (3) Explain reasonable reasons why novel synthetic fatty acid esters were selected among FAHFAs in this study. As well, why did the authors investigate their antioxidant potential? (4) This study demonstrated Nrf2 accumulation in nuclei and induction of Nrf2-target genes in cells treated with 12-EPAHSA, but not investigated “Nrf2 translocation from cytoplasm to nucleus” at all. The sentences describing the translocation (Line 22, 127, 243, 280, and 297) must be modified appropriately. Otherwise, cytoplasmic Nrf2 should be detected in cells before and after 12-EPAHSA supplementation. (5) To show the Nrf2 dependency of the FAHFA-mediated antioxidant response, expression of the Nrf2-target genes must be examined in 12-EPAHSA-treated cells, in which Nrf2 expression levels are reduced. (6) Regarding Figure 3- (6-1) It is generally considered that newly synthesized Nrf2 avoids capture by Keap1 and accumulates in nuclei under oxidative stress conditions. In this regard, the cartoon showing nuclear translocation of Nrf2 released from Keap1 capturing is incorrect. (6-2) It is generally considered that electrophiles cut off the Keap1-Nrf2 interaction through adducting Keap1. Do the authors think that FAHFAs directly stimulate Nrf2 as shown in the schema? Please propose the authors’ hypothesis on how FAHFAs intervene in the Keap1-Nrf2 interaction. Is it possible to think that FAHFAs inactivate Keap1 as electrophiles? (6-3) “IRON” and “HEME” should be “Iron” and “Heme”, respectively. (7) Because the data in the supplementary information are important for this manuscript, which has enough room, include them in the regular figures. Minor points (8) Since “nuclear factor (erythroid-derived 2)-like 2 (Nrf2) (Line 14)” indicates the name of the gene for Nrf2 (NFE2L2), it should be “nuclear factor erythroid 2-related factor 2 (Nrf2)”. (9) “NQ1” should be “NQO1” in Lines 24 and 290. (10) Regarding the term “phase II antioxidant enzymes (Line 24, 54, 274, 286, and 290)”, are there phase I antioxidant enzymes? This term is inappropriate because phase I and II indicate detoxification phases governed by AhR (P450 enzymes) and Nrf2 (glutathione transferases), respectively. Therefore, “phase II enzymes” does NOT include SOD1, HO1, and CAT among the listed genes in this manuscript. (11) Line 37 “acid (PAHSA)” should be “acids (PAHSAs)” (12) Insert “a” between “in” and “Keap1-dependent” in Line 49. (13) “Proteomic levels” may be “Protein levels” correctly in Line 235. (14) “western blot” is “Western blot” (Line 236). (15) Where they describe HepG2 cells (Line 309), the authors should excuse that the C3A cell line used in this manuscript is a subclone of HepG2 cells. (16) Appropriate citations are required for the sentence “Nrf2 targeted cytoprotective genes such as NQO1, GCLM, GCLC, SOD-1, CAT, and HO-1 (Line 250)” (17) “linolenic (LA)-treated” should be “linoleic acid (LA)-treated” or “linolenate (LA)-treated” (Line 254).

Author Response

Reviewer 2

The authors were intended to elucidate effects of branched fatty acid esters of hydroxy fatty acids (FAHFAs), which are endogenous lipids expected to protect tissues from various damages, on cellular mechanisms of Nrf2-mediated antioxidant response. As the result, they found that a FAHFA, eicosapentaenoic acid esterified 12-hydroxy stearic acid 19 (12-EPAHSA), stimulates nuclear accumulation of Nrf2 protein and expression of Nrf2-target genes related to antioxidant response. Based on the results, they insist that FAHFAs are cytoprotective from oxidative stress via activating the master antioxidative transcription factor Nrf2. However, the causal nexus between FAHFAs and the Nrf2 pathway is unclear due to lack of mechanism analyses. To complete this study for publication, additional experiments are strongly advocated as below.

Response: Thank you for your valuable comments.

Major points

  • Keap1 inactivates Nrf2 while Nrf2 activates antioxidant gene expression. Therefore, “Activation of Keap1-Nrf2 Signaling” and “Keap1-Nrf2 activation” is inaccurate and confusing (In the title; Line 21, 28, 48, 228, 241, 273, 275, and 293). Additionally, this study did not investigate roles of Keap1 at all. These phrases should be replaced with “activation of Nrf2” or “Nrf2 activation”.

Response: Thank you for your valuable comments. Throughout the manuscript “activation of Nrf2” or “Nrf2 activation terms were used as suggested.

  • This study used only one cell line (C3A). However, the effects of 12-EPAHSA must be confirmed by additional experiments using multiple cell lines.

Response: Thank you for your suggestions, Since FAHFAs are earlier detected in liver and have ability to enhance insulin sensitivity (Yore M etal Cell 2014), we mainly focused on the Human liver derived cells (C3A) in the present work. Moreover, liver metabolizes xenobiotics and endogenous molecules to maintain metabolic homeostasis, where oxidative stress plays a crucial role for liver function.

  • Explain reasonable reasons why novel synthetic fatty acid esters were selected among FAHFAs in this study. As well, why did the authors investigate their antioxidant potential?

Response: Thank you for your comments. The FAHFAs are synthesized from each class of monounsaturated fatty acids (MUFA) and poly unsaturated fatty acids (PUFA), to evaluate their antioxidant potential. The lines 54-57 described why we investigated their antioxidant potential. Moreover, PUFAs are reported to be potent antioxidants (Zgórzyńska etal , 2017), but there are no reports on their derivatives such as FAHFAs as antioxidants.

  • This study demonstrated Nrf2 accumulation in nuclei and induction of Nrf2-target genes in cells treated with 12-EPAHSA, but not investigated “Nrf2 translocation from cytoplasm to nucleus” at all. The sentences describing the translocation (Line 22, 127, 243, 280, and 297) must be modified appropriately. Otherwise, cytoplasmic Nrf2 should be detected in cells before and after 12-EPAHSA supplementation.

Response: Thank you for your suggestions. We modified the sentences as reviewer suggested.

Line 22 is modified as “To define their mode of action relative levels of nuclear fraction Nrf2 was determined and found the higher amount of Nrf2 in nucleus of cells treated with 12-EPAHSA compare to the control” Please refer line 21-23 in revised MS.

Line 127 is modified as “The nuclear fraction Nrf2 accumulation is determined by the nuclear protein extraction kit and Western blotting analysis” Please refer line 123 -124 in the revised MS.

Line 243 is modified as “The effect of the compound 12-EPAHSA (4) on the accumulation of Nrf2 protein in nuclear fraction was examined using nuclear protein extraction kit” Please refer line 240-241 in the revised MS.

Line 280 is modified as “the accumulation of higher amount of Nrf2 in nucleus of cells treated with 12-EPAHSA compare to the control suggesting the possible activation of nuclear Nrf2. Please refer line 291-291 in the revised MS.

Line 297 is modified as “12-EPAHSA could activate the nuclear fraction Nrf2 and induce the activation of Nrf2 targeted genes to protect cells against oxidative stress related damages” Please refer line 302-303 in the revised MS.

  • To show the Nrf2 dependency of the FAHFA-mediated antioxidant response, expression of the Nrf2-target genes must be examined in 12-EPAHSA-treated cells, in which Nrf2 expression levels are reduced.

Response: Thank you for your suggestions. It is the first report to show the FAHFAs as Nrf2 activators. The relative amount of Nrf2 is certainly low in C3A cells (as seen in control experiments) and all the genes we examined are known to be Nrf2 target genes (NQO1, GCLM, GCLC, SOD-1, CAT, and HO-1), which were activated by accumulation of nuclear Nrf2 and their response were concentration dependent as shown in Fig. 1E.

  • Regarding Figure 3- (6-1) It is generally considered that newly synthesized Nrf2 avoids capture by Keap1 and accumulates in nuclei under oxidative stress conditions. In this regard, the cartoon showing nuclear translocation of Nrf2 released from Keap1 capturing is incorrect.

Response: We agree with the reviewer. However, there may studies suggests that accumulation of nuclear Nrf2 is due to the Nrf2 translocation from cytoplasm (Sharath babu GR etal, Front. Pharmacol 2017, Namani A etal BBA Mol. Cell Res. 2014, Joko S etal 2017, Fuda H etal 2019). Hence, we have shown the translocation in cartoon as a “plausible pathway”.

(6-2) It is generally considered that electrophiles cut off the Keap1-Nrf2 interaction through adducting Keap1. Do the authors think that FAHFAs directly stimulate Nrf2 as shown in the schema? Please propose the authors’ hypothesis on how FAHFAs intervene in the Keap1-Nrf2 interaction. Is it possible to think that FAHFAs inactivate Keap1 as electrophiles?

Response: We assume that oxidized products of FAHFAs could be electrophilic and destabilize the interaction between Keap-Nrf2. Because in a previous study, it was shown that PUFA oxidation products activate the Nrf2 via such interactions (Gao L etal J. Biol Chem 2006, 282, 2529). We have modified our schema showing not the direct stimulation, possibly involving multi-step interaction of electrophilic products to destabilize Keap1-Nrf2.

(6-3) “IRON” and “HEME” should be “Iron” and “Heme”, respectively.

Response: Corrected.

  • Because the data in the supplementary information are important for this manuscript, which has enough room, include them in the regular figures.

Response: Thank you for your suggestions. We have moved Table s1 (as text) and Figure s3 (as Figure 3) to the main manuscript.

Minor points

  • Since “nuclear factor (erythroid-derived 2)-like 2 (Nrf2) (Line 14)” indicates the name of the gene for Nrf2 (NFE2L2), it should be “nuclear factor erythroid 2-related factor 2 (Nrf2)”.

Response: corrected.

  • “NQ1” should be “NQO1” in Lines 24 and 290.

Response: corrected.

  • Regarding the term “phase II antioxidant enzymes (Line 24, 54, 274, 286, and 290)”, are there phase I antioxidant enzymes? This term is inappropriate because phase I and II indicate detoxification phases governed by AhR (P450 enzymes) and Nrf2 (glutathione transferases), respectively. Therefore, “phase II enzymes” does NOT include SOD1, HO1, and CAT among the listed genes in this manuscript.

Response: Thank you for your comments. The term “Phase II” was deleted throughout and kept term “antioxidant enzymes”.

  • Line 37 “acid (PAHSA)” should be “acids (PAHSAs)”

Response: Corrected.

  • Insert “a” between “in” and “Keap1-dependent” in Line 49.

Response: Corrected.

(13) “Proteomic levels” may be “Protein levels” correctly in Line 235.

Response: Corrected.

 (14) “western blot” is “Western blot” (Line 236).

Response: Corrected.

(15) Where they describe HepG2 cells (Line 309), the authors should excuse that the C3A cell line used in this manuscript is a subclone of HepG2 cells.

Response: Yes, we agree we mentioned that in line 107 in the revised manuscript.

 (16) Appropriate citations are required for the sentence “Nrf2 targeted cytoprotective genes such as NQO1, GCLM, GCLC, SOD-1, CAT, and HO-1 (Line 250)”

Response: Line 250 is explaining our experimental results, the evidence for our results consistent with previous studies we already cited reference 10 and 17 in the discussion section.

 (17) “linolenic (LA)-treated” should be “linoleic acid (LA)-treated” or “linolenate (LA)-treated” (Line 254).

Response: Corrected.

Reviewer 3 Report

Gowda et al. describe an effect of eicosapentaenoic acid (EPA) esters of hydroxy fatty acids (so called FAHFAs, 12-EPAHSA) on Nfe2l2 signaling. The Nfe2l2 pathway is able to resolve oxidative stress responses. The authors outline the synthesis and analysis of the EPA-FAHFAs. They show that EPA-FAHFAs activate the Nfe2l2 pathway and have effects on lipid droplet peroxidation in cultured cells.

Major points:

  1. The effect of the unesterified components of 12-EPAHSA are not presented as controls (e.g. EPA or stearic acid treatment alone), which would strengthen the argument that the actual FAHFA is the relevant substance for inducing Nfe2l2 signaling. EPA and stearic acid themselves are stressful agents. The explanation why EPA was not tested is not conclusive, as the activation of luciferase is higher than during treatment with 12-EPAHSA in concentrations relative to the respective IC50. It is reasonable to assume that treatment with FAHFAs causes cellular stress which itself can alter gene expression. Therefore, it would be interesting to determine cellular stress markers (e.g. via qPCR CHOP/Ddit3)
  2. The authors claim that the effect of FAHFAs on the cell are beneficial because they are antioxidants, even though this is never shown in the paper. If anything, they act as oxidants as they engage the Nfe2l2 pathway. An more detailed analysis of 12-EPAHSA treatment on oxidative stress (e.g. upon H2O2 treatment) would give more insight into that. Also, the immunofluorescence analysis is a start but a more quantitative analysis fo oxidative stress is required.
  3. It is unclear whether the effect on Nfe2l2 target genes described in Figure 1E is actually mediated by Nfe2l2, e.g. knockdown of Nfe2l2 via siRNA would prove the causality. Also, the ARE reporter is not specific for Nfe2l2 but also reports Nfe2l1 activation.

Minor Points:

  1. English language can be improved
  2. The differentiation between small and big lipid droplets seems arbitrary and the analysis is unclear.
  3. The introduction claims that the role of FAHFAs on their possible biosynthesis in liver cells was interrogated. Besides a figure in the supplementary data this is not in the paper. It would be appreciated if the authors presented more insights into that. Please clarify.
  4. Cat.Nr. for Anti-Lamin is given but not for Anti-Nfe2l2

Author Response

Reviewer 3:

Gowda et al. describe an effect of eicosapentaenoic acid (EPA) esters of hydroxy fatty acids (so called FAHFAs, 12-EPAHSA) on Nfe2l2 signaling. The Nfe2l2 pathway is able to resolve oxidative stress responses. The authors outline the synthesis and analysis of the EPA-FAHFAs. They show that EPA-FAHFAs activate the Nfe2l2 pathway and have effects on lipid droplet peroxidation in cultured cells.

Response: Thank you for your valuable comments.

Major points:

The effect of the unesterified components of 12-EPAHSA are not presented as controls (e.g. EPA or stearic acid treatment alone), which would strengthen the argument that the actual FAHFA is the relevant substance for inducing Nfe2l2 signaling. EPA and stearic acid themselves are stressful agents. The explanation why EPA was not tested is not conclusive, as the activation of luciferase is higher than during treatment with 12-EPAHSA in concentrations relative to the respective IC50. It is reasonable to assume that treatment with FAHFAs causes cellular stress which itself can alter gene expression. Therefore, it would be interesting to determine cellular stress markers (e.g. via qPCR CHOP/Ddit3)

Response: Thank you for your comments. Our aim is to check the activity of FAHFAs, not free fatty acids, as already reported polyunsaturated fatty acids can acts as antioxidant via Nrf2 signaling in rat astrocytes (Reference 19 of main MS). At first, we have screened the FAHFAs for their Nrf2 activation and found that only EPA derived FAHFAs are active not LA or OA derived. Which further lead us to think which structural part of FAHFA is active, hence we have tested each fatty acid individually by reporter gene assay and found that EPA could be the active part not 12-HSA. It’s of note that, EPA showed more toxicity compare to 12-EPAHSA. Hence, FAHFAs are more promising candidates than their free fatty acids.

We performed the qPCR analysis of CHOP/Ddit3 an oxidative stress biomarker and added the data in supporting information Figure S2. 12-EPAHSA significantly reduced the CHOP/Ddit3 levels at 32 µM, however the effect was not concentration dependent. These lines are added in the revised MS please refer line 249-251.

The authors claim that the effect of FAHFAs on the cell are beneficial because they are antioxidants, even though this is never shown in the paper. If anything, they act as oxidants as they engage the Nfe2l2 pathway. An more detailed analysis of 12-EPAHSA treatment on oxidative stress (e.g. upon H2O2 treatment) would give more insight into that. Also, the immunofluorescence analysis is a start but a more quantitative analysis fo oxidative stress is required.

Response: Thank you for your suggestions. It is evident in the literature that when cells exposed to oxidative stress, lipid droplets (LD) will accumulate (Jarc E etal 2019, Yale J Boil Med, 92, 435) to protect membranes from peroxidation reactions. In our study, we analysed the normal and oxidized lipid droplets by the method originally established in our lab and shown that 12-EPAHSA can inhibit the oxidation of small LDs concentration dependently. These results could support our claim that FAHFAs are act as antioxidants, Moreover FAHFAs are not small molecules to be act as direct antioxidants, hence we evaluated their possible activity as antioxidants via Nrf2 regulation.

It is unclear whether the effect on Nfe2l2 target genes described in Figure 1E is actually mediated by Nfe2l2, e.g. knockdown of Nfe2l2 via siRNA would prove the causality. Also, the ARE reporter is not specific for Nfe2l2 but also reports Nfe2l1 activation.

Response: Thank you for your comments. There are many reports showing that activation of Nrf2-regulated genes is associated with translocation of Nrf2 to nuclear fraction (Nguyen T etal 2005, Joko S etal 2017, Fuda H etal 2019, Tonelli C 2018, and Michaeloudes C etal, 2020). Hence, the observed increase of Nrf2 induced by 12-EPAHSA (Fig. 1D) supports a possible role for FAHFAs to activate Nrf2-regulated genes. In addition, the knockdown of Nrf1 can increase the expression of antioxidant genes including NQO1 and GSTP1 (Ohtsuji M etal, 2008, J Biol Chem 283, 33554). Moreover, activation of NQO1 is induced by Nrf2 much more strongly than that by Nrf1Δ30 in a reporter gene assay system (Ohtsuji M etal, 2008, J Biol Chem 283, 33554). Our assay is based on a similar luciferase reporter gene assay system. Considering the above all, we assume that the translocated nuclear Nrf2 rather than Nrf1 plays a major role in the FAHFAs-induced expression of antioxidant enzymes. These lines were added from 295-299 in the revised MS. However, the reviewer’s comment is quite useful to elucidate precise mechanism for antioxidative effect of FAHFAs.

Minor Points:

English language can be improved

Response: Thank you for your suggestions. We have improved the English language wherever necessary in the revised manuscript.

The differentiation between small and big lipid droplets seems arbitrary and the analysis is unclear.

Response: Since the method of analysis of small and big lipid droplets is already reported by our group, we haven’t focused in detail on the analysis. We have cited the reference 18 for the detailed analysis.

The introduction claims that the role of FAHFAs on their possible biosynthesis in liver cells was interrogated. Besides a figure in the supplementary data this is not in the paper. It would be appreciated if the authors presented more insights into that. Please clarify.

Response: Thank you for your suggestions. We have moved that Figure S2 to the main text, as Figure 4. A few lines were added concerning this data, see lines 268-274 and 318-322.

Cat.Nr. for Anti-Lamin is given but not for Anti-Nfe2l2

Response: catalogue no. was added, see line 128.

Round 2

Reviewer 2 Report

The authors partially addressed to my comments. Major concerns still remain in the revised manuscript. (Comment-1) Keap1 inactivates Nrf2 while Nrf2 activates antioxidant gene expression. Therefore, “Activation of Keap1-Nrf2 Signaling” and “Keap1-Nrf2 activation” is inaccurate and confusing (In the title; Line 21, 28, 48, 228, 241, 273, 275, and 293). Additionally, this study did not investigate roles of Keap1 at all. These phrases should be replaced with “activation of Nrf2” or “Nrf2 activation”. (Response to Comment-1) Thank you for your valuable comments. Throughout the manuscript “activation of Nrf2” or “Nrf2 activation terms were used as suggested. (New Comment-1) Among the 9 problematic points listed in the original comment, one point (Line 241 in the original manuscript) has not been corrected. The authors should carefully discuss their interpretation on roles of Keap1 in FAHFA-mediated antioxidant response, which they have not addressed in this study at all. (Comment-2) This study used only one cell line (C3A). However, the effects of 12-EPAHSA must be confirmed by additional experiments using multiple cell lines. (Response to Comment-2) Thank you for your suggestions, Since FAHFAs are earlier detected in liver and have ability to enhance insulin sensitivity (Yore M etal Cell 2014), we mainly focused on the Human liver derived cells (C3A) in the present work. Moreover, liver metabolizes xenobiotics and endogenous molecules to maintain metabolic homeostasis, where oxidative stress plays a crucial role for liver function. (New Comment-2) This comment was ignored. Additional experiments using hepatic cell lines other than C3A must be conducted. (Comment-3) Explain reasonable reasons why novel synthetic fatty acid esters were selected among FAHFAs in this study. As well, why did the authors investigate their antioxidant potential? (Response to Comment-3) Thank you for your comments. The FAHFAs are synthesized from each class of monounsaturated fatty acids (MUFA) and poly unsaturated fatty acids (PUFA), to evaluate their antioxidant potential. The lines 54-57 described why we investigated their antioxidant potential. Moreover, PUFAs are reported to be potent antioxidants (Zgórzyńska etal , 2017), but there are no reports on their derivatives such as FAHFAs as antioxidants. (New Comment-3) Please describe whether 12-EPAHSA and 12-EPAHOA, the “novel synthetic fatty acid esters“ used here, are endogenous lipids or not. (Comment-4) This study demonstrated Nrf2 accumulation in nuclei and induction of Nrf2-target genes in cells treated with 12-EPAHSA, but not investigated “Nrf2 translocation from cytoplasm to nucleus” at all. The sentences describing the translocation (Line 22, 127, 243, 280, and 297) must be modified appropriately. Otherwise, cytoplasmic Nrf2 should be detected in cells before and after 12-EPAHSA supplementation. (Response to Comment-4) Thank you for your suggestions. We modified the sentences as reviewer suggested. Line 22 is modified as “To define their mode of action relative levels of nuclear fraction Nrf2 was determined and found the higher amount of Nrf2 in nucleus of cells treated with 12-EPAHSA compare to the control” Please refer line 21-23 in revised MS. Line 127 is modified as “The nuclear fraction Nrf2 accumulation is determined by the nuclear protein extraction kit and Western blotting analysis” Please refer line 123 -124 in the revised MS. Line 243 is modified as “The effect of the compound 12-EPAHSA (4) on the accumulation of Nrf2 protein in nuclear fraction was examined using nuclear protein extraction kit” Please refer line 240-241 in the revised MS. Line 280 is modified as “the accumulation of higher amount of Nrf2 in nucleus of cells treated with 12-EPAHSA compare to the control suggesting the possible activation of nuclear Nrf2. Please refer line 291-291 in the revised MS. Line 297 is modified as “12-EPAHSA could activate the nuclear fraction Nrf2 and induce the activation of Nrf2 targeted genes to protect cells against oxidative stress related damages” Please refer line 302-303 in the revised MS. (New Comment-4) The phrase “nuclear fraction Nrf2” may be replaced with “nuclear Nrf2”. “Nrf2 accumulation is determined … (Line 123)” should be “Nrf2 accumulation was determined …” (Comment-5) To show the Nrf2 dependency of the FAHFA-mediated antioxidant response, expression of the Nrf2-target genes must be examined in 12-EPAHSA-treated cells, in which Nrf2 expression levels are reduced. (Response to Comment-5) Thank you for your suggestions. It is the first report to show the FAHFAs as Nrf2 activators. The relative amount of Nrf2 is certainly low in C3A cells (as seen in control experiments) and all the genes we examined are known to be Nrf2 target genes (NQO1, GCLM, GCLC, SOD-1, CAT, and HO-1), which were activated by accumulation of nuclear Nrf2 and their response were concentration dependent as shown in Fig. 1E. (New Comment-5) This comment was ignored. To demonstrate the authors’ conclusion that FAHFAs are antioxidants via Nrf2 activation, 12-EPAHSA-inducible Nrf2-target gene expression must be examined in Nrf2-knockdown (or knockout) cells. • (Comment-6-1) Regarding Figure 3, It is generally considered that newly synthesized Nrf2 avoids capture by Keap1 and accumulates in nuclei under oxidative stress conditions. In this regard, the cartoon showing nuclear translocation of Nrf2 released from Keap1 capturing is incorrect. (Response: to Comment-6-1) We agree with the reviewer. However, there may studies suggests that accumulation of nuclear Nrf2 is due to the Nrf2 translocation from cytoplasm (Sharath babu GR etal, Front. Pharmacol 2017, Namani A etal BBA Mol. Cell Res. 2014, Joko S etal 2017, Fuda H etal 2019). Hence, we have shown the translocation in cartoon as a “plausible pathway”. (New Comment-6-1) Once Keap1 captures Nrf2, Keap1 never releases Nrf2. Nuclear Nrf2 is derived from newly synthesized Nrf2 that escapes Keap1-mediated degradation, but not from Keap1-binding Nrf2. Therefore, the arrow between the Keap1-Nrf2 complex and cytoplasmic free Nrf2 is incorrect. (Comment-6-2) It is generally considered that electrophiles cut off the Keap1-Nrf2 interaction through adducting Keap1. Do the authors think that FAHFAs directly stimulate Nrf2 as shown in the schema? Please propose the authors’ hypothesis on how FAHFAs intervene in the Keap1-Nrf2 interaction. Is it possible to think that FAHFAs inactivate Keap1 as electrophiles? (Response to Comment-6-2) We assume that oxidized products of FAHFAs could be electrophilic and destabilize the interaction between Keap-Nrf2. Because in a previous study, it was shown that PUFA oxidation products activate the Nrf2 via such interactions (Gao L etal J. Biol Chem 2006, 282, 2529). We have modified our schema showing not the direct stimulation, possibly involving multi-step interaction of electrophilic products to destabilize Keap1-Nrf2. (New Comment-6-2) As the authors responded, contribution of this study to this schema is very small. Because they have not conducted functional analyses of Nrf2, Keap1, and Nrf2-target gene products, the confusing schema is unhelpful for comprehension of the outcome from this study. I think that the schema may be replaced with the graphical abstract (Page 1) after minor modification.

Author Response

Response letter was attached

Reviewer 3 Report

I have to say I am disappointed by the authors response to the criticism of their work. Criticism is the origin of progress.  If the authors to continue to reject scientifically based criticism the authors will not be able to improve their work now and in the future.

My points were directed at the interpretation and conclusions of the authors. Furthermore, there is large overlap in the points raised by the other reviewer #2 and mine.

The remaining major points are:

  1. The authors claim that the effect of FAHFAs on the cell are beneficial because they are antioxidants, even though this is never shown in the paper. If anything, they act as oxidants as they engage the Nfe2l2 pathway. An more detailed analysis of 12-EPAHSA treatment on oxidative stress (e.g. upon H2O2 treatment) would give more insight into that. The immunofluorescence analysis is no direct measure of oxidative stress - it only shows lipid droplets. The authors CANNOT make these conclusions.
  2. It is unclear whether the effect on Nfe2l2 target genes described in Figure 1E is actually mediated by Nfe2l2, e.g. knockdown of Nfe2l2 via siRNA would prove the causality. Also, the ARE reporter is not specific for Nfe2l2 but also reports Nfe2l1 activation. The authors CANNOT claim FAHSAs are antioxidative via Nrf2 activation - NO CAUSALITY is proven in the paper, more work is needed for that.

Author Response

Response letter attached.

Round 3

Reviewer 2 Report

Nrf2 knockdown (or knockout) experiments are mandatory for this manuscript. This manuscript is uncompleted as a scientific article and fails to give enough impact without the experiments. This point also concerns another reviewer. Again, analyses of multiple cell lines are also required. The HepG2 cell line does not satisfy because the C3A cell line is its derivative. Figure 4 is still confusing. Does 12-EPAHSA stimulate Nrf2 activation and Nrf2 nuclear accumulation independently? Most part of the figure illustrates function of Nrf2-target gene products (Heme degradation, glutathione synthesis, and quinone metabolism), which were not addressed in this study. “HEME” and “IRON” went back to the former incorrect forms. I repetitively suggest that the schema should be replaced with the graphical abstract.